# LDA-AQU: Adaptive Query-guided Upsampling via Local Deformable Attention

Zewen Du
Beijing Institute of Technology
Beijing, China
3120220818@bit.edu.cn

Zhenjiang Hu
Beijing Institute of Technology
Beijing, China
3220220769@bit.edu.cn

Guiyu Zhao
Beijing Institute of Technology
Beijing, China
3120220906@bit.edu.cn

Ying Jin
Beijing Institute of Technology
Beijing, China
jinyinghappy@bit.edu.cn

Hongbin Ma*
Beijing Institute of Technology
Beijing, China
mathmhb@bit.edu.cn

## Abstract

Feature upsampling is an essential operation in constructing deep convolutional neural networks. However, existing upsamplers either lack specific feature guidance or necessitate the utilization of high-resolution feature maps, resulting in a loss of performance and flexibility. In this paper, we find that the local self-attention naturally has the feature guidance capability, and its computational paradigm aligns closely with the essence of feature upsampling (*i.e.*, feature reassembly of neighboring points). Therefore, we introduce local self-attention into the upsampling task and demonstrate that the majority of existing upsamplers can be regarded as special cases of upsamplers based on local self-attention. Considering the potential semantic gap between upsampled points and their neighboring points, we further introduce the deformation mechanism into the upsampler based on local self-attention, thereby proposing LDA-AQU. As a novel dynamic kernel-based upsampler, LDA-AQU utilizes the feature of queries to guide the model in adaptively adjusting the position and aggregation weight of neighboring points, thereby meeting the upsampling requirements across various complex scenarios. In addition, LDA-AQU is lightweight and can be easily integrated into various model architectures. We evaluate the effectiveness of LDA-AQU across four dense prediction tasks: object detection, instance segmentation, panoptic segmentation, and semantic segmentation. LDA-AQU consistently outperforms previous state-of-the-art upsamplers, achieving performance enhancements of 1.7 AP, 1.5 AP, 2.0 PQ, and 2.5 mIoU compared to the baseline models in the aforementioned four tasks, respectively.

## CCS Concepts

• **Computing methodologies** → **Object detection**; **Image segmentation**.

*Corresponding author.

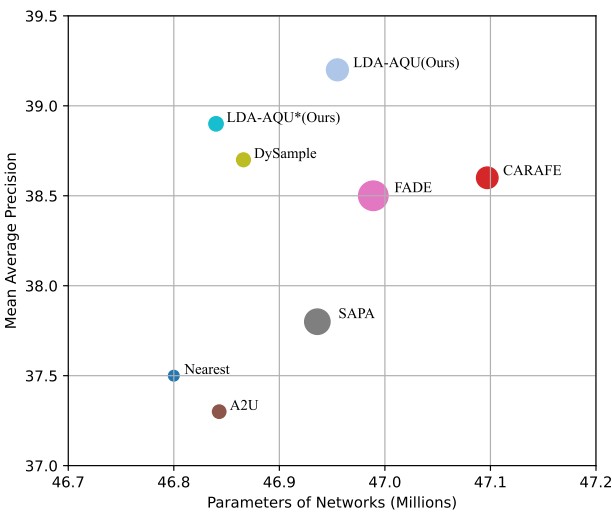

**Figure 1: Comparison of various upsamplers in terms of network parameters, Mean Average Precision (mAP) and FLOPs (indicated by area of circles) using Faster R-CNN [31] with ResNet-50 [12] as the baseline model.**

## Keywords

Feature upsampling, Local self-attention, Local deformable attention, Dynamic upsampler, Dense prediction tasks

**ACM Reference Format:**
Zewen Du, Zhenjiang Hu, Guiyu Zhao, Ying Jin, and Hongbin Ma. 2024. LDA-AQU: Adaptive Query-guided Upsampling via Local Deformable Attention. In *Proceedings of the 32nd ACM International Conference on Multimedia (MM '24), October 28-November 1, 2024, Melbourne, VIC, Australia.* ACM, New York, NY, USA, 9 pages. https://doi.org/10.1145/3664647.3680789

## 1 Introduction

As a fundamental operator in deep convolutional neural networks, feature upsampling is widely utilized in various dense prediction tasks, including object detection, semantic segmentation, and image inpainting, *etc.* Given the spatial downsampling characteristics of convolutional and pooling operators, feature upsampling emerges as a crucial inverse operation, indispensable for meeting task-specific requirements. For instance, it facilitates the restoration

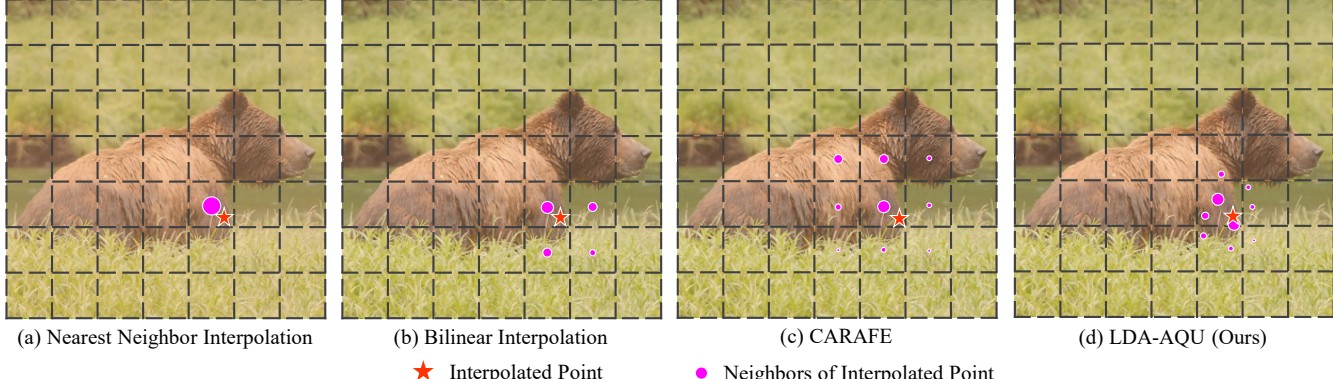

| (a) Nearest Neighbor Interpolation | (b) Bilinear Interpolation | (c) CARAFE | (d) LDA-AQU (Ours) |

★ Interpolated Point          ● Neighbors of Interpolated Point

**Figure 2: The difference in neighboring point selection schemes between LDA-AQU and other widely used upsamplers. Given an upsampled point (red star), LDA-AQU employs the query-guided mechanism to predict the deformation offset and aggregation weight of neighboring points, enabling adaptation to upsampling tasks across multiple scales.**

of spatial resolution in pixel-level dense prediction tasks and enables multi-scale feature fusion in feature pyramid network (FPN) [18].

Commonly used upsamplers, such as Nearest Neighbor Interpolation and Bilinear Interpolation, aggregate features from neighboring points in a manually designed paradigm, making it difficult to address the requirements of various upsampling tasks simultaneously. Subsequently, several learnable upsampling methods have been proposed, including deconvolution [23] and Pixel Shuffle [32], *etc.* However, these methods typically learn a fixed set of parameters for the upsampling kernel, applying the same operation to all spatial positions of the input feature map, leading to suboptimal upsampling results. To enhance the dynamic adaptability of the upsampling operator and enable it to address various upsampling tasks in complex scenes, dynamic filter-based upsamplers have been proposed [26, 35, 41]. However, these methods either lack specific feature guidance or require the intervention of high-resolution images, limiting their application scenarios and performance.

Most existing upsampling operators can be viewed as a weighted aggregation of features within local neighborhoods surrounding upsampled points (*i.e.*, feature reassembly). We observe that this is consistent with local self-attention, which determines attention weights and extracts contextual information from uniform neighboring points. However, local self-attention naturally incorporates the query-guided mechanism, aligning well with the upsampling task. This involves adaptively aggregating neighborhood features based on the attributes of the upsampled points, resulting in explicit point affiliations [26]. These insights inspire us to integrate local self-attention into the upsampling task, achieving adaptive upsampling with query guidance in a single layer.

In this paper, we introduce a method for incorporating local self-attention into feature upsampling tasks. Additionally, we note that using fixed, uniform neighboring points may lead to suboptimal upsampling result. As depicted in Figure 2, uniform neighboring point selection in feature maps with high downsampling strides may result in notable semantic disparities, hindering high-resolution feature map generation. To address this, we introduce the deformation mechanism to dynamically adjust the positions of neighboring points based on the features of query points (*i.e.*, upsampled points) and their contextual information, aiming to further enhance the

model's adaptability. Based on above, we have named our method as LDA-AQU, which offers the following advantages compared to other dynamic upsamplers: 1) operates on a single layer without requiring high-resolution inputs; 2) possesses query-guided capability, enabling the interactive generation of dynamic upsampling kernels using the features of the query points and their neighboring points; 3) exhibits local deformation capability, permitting dynamic adjustment of positions to neighboring points based on the contextual information of query points. These properties enable LDA-AQU to achieve superior performance while remaining lightweight.

Through extensive experiments conducted on four dense prediction tasks including object detection, semantic segmentation, instance segmentation, and panoptic segmentation, we have validated the effectiveness of LDA-AQU. For instance, LDA-AQU can obtain +1.7 AP gains for Faster R-CNN [31], +1.5 AP gains for Mask R-CNN [11], +2.0 PQ gains for Panoptic FPN [14] on MS COCO dataset [19]. On the semantic segmentation task, LDA-AQU also brings +2.5 mIoU gains for UperNet [36] on ADE20K dataset [40]. On the aforementioned four dense prediction tasks, our LDA-AQU consistently outperforms the previous state-of-the-art upsampler while maintaining a similar FLOPs and parameters.

## 2  Related Work

### 2.1  Feature Upsampling

Commonly used upsamplers, such as Nearest Neighbor Interpolation and Bilinear Interpolation, are popular for their simplicity and efficiency in various visual tasks. To enhance the adaptability, some learnable upsamplers have been proposed. Deconvolution [23] employs a reverse convolution operation to achieve the upsampling of feature maps. Pixel Shuffle (PS) [32] increases the spatial resolution of the feature map by shuffling features along both spatial and channel directions. With the popularity of dynamic networks [3, 38, 41], some upsamplers based on dynamic kernels have been introduced. CARAFE [35] utilizes a content-aware approach to generate dynamic aggregate weights. IndexNet [24] models various upsampling operators as different index functions and proposes several index networks to generate indexes for guiding upsampling. SAPA [26] introduces the concept of point membership into feature upsampling

and guides kernel generation through the similarity between semantic clusters. In comparison, our LDA-AQU utilizes the features of the query point to generate deformed offsets and aggregate weights of neighboring points, achieving dynamic feature upsampling with feature guidance from a single low-resolution input.

## 2.2 Dense Prediction Tasks

Dense prediction involves pixel-level tasks such as object detection [16, 30, 31], instance segmentation [1, 9, 11], panoptic segmentation [14, 15, 17], and semantic segmentation [4, 23, 37]. Models for such tasks typically include a backbone network [12, 13, 33], a feature pyramid network [10, 18, 20], and one or more task heads. The backbone network reduces data dimensions and extracts salient features to decrease computational complexity and capture robust semantic information. The feature pyramid network connects multi-scale features to enhance the model's perception across scales. The task head links extracted features to the prediction task, serving as the primary distinction among different models.

For instance, Faster R-CNN [31] uses a detection head for object recognition and localization tasks. Mask R-CNN [11] achieves both object detection and instance segmentation by adding an instance segmentation head based on Faster R-CNN. Similarly, Panoptic FPN [14] integrates semantic segmentation, instance segmentation, and panoptic segmentation tasks by incorporating an additional semantic segmentation head based on Mask R-CNN. UperNet [36] uniformly conducts scene perception and parsing by integrating various task heads. Due to the spatial reduction characteristics of convolution and pooling operations, feature upsampling becomes essential for accomplishing above dense prediction tasks. Our LDA-AQU can be easily integrated into these frameworks and consistently brings stable performance improvements.

## 2.3 Vision Transformer

As vanilla self-attention in Vision Transformer (ViT) [8] capture global contextual information through dense interactions, the computational complexity and memory usage become unbearable for high-resolution inputs. Therefore, in recent years, numerous studies have aimed to optimize the efficiency of ViT. Swin Transformer [22] decreases interactive tokens by confining them to non-overlapping windows, and then captures global dependencies through window sliding. CSwin Transformer [7] further improves efficiency with a cross-shaped interactive window. Additionally, some methods based on local self-attention [27, 29, 34, 39] have been proposed to optimize model efficiency and introduce the inductive bias of convolution. We incorporate local self-attention into the upsampling task and introduce the deformation mechanism for neighboring points to enhance the model's dynamic adaptability.

## 3 Method

First, we will provide a brief overview of self-attention and local self-attention. Then, we will elaborate on our approach for extending the local self-attention to address feature upsampling tasks, which we refer to as LA-AQU. Finally, we will introduce the integration of the deformation mechanism into LA-AQU, presenting LDA-AQU as a means to enhance its adaptability in complex scenarios.

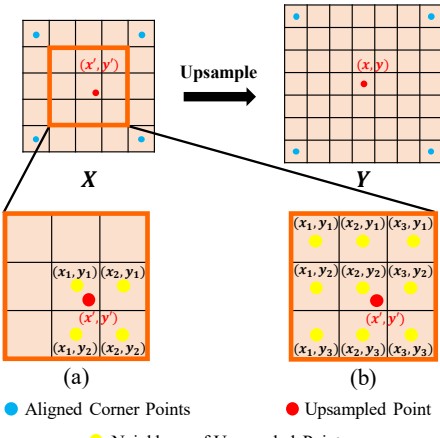

**Figure 3: The upsampling kernels of Bilinear Interpolation-Based (a) and Local Self-Attention-Based Upsamplers (b).**

## 3.1 Preliminary

*3.1.1 Self-Attention.* Given a flattened input feature map $x \in \mathbb{R}^{N \times C}$, where $C$ is the channel size and $N = H \times W$ is the number of tokens along the spatial dimension, the output of the $i$-th token $z_i$ after standard self-attention can be expressed as:

$$(q, k, v) = (xW^q, xW^k, xW^v) \tag{1}$$

$$z_i = \sum_{j=1}^{N} \frac{\exp(q_i k_j^{\mathrm{T}})}{\sum_{m=1}^{N} \exp(q_i k_m^{\mathrm{T}})} v_j \tag{2}$$

where $W^q, W^k, W^v \in \mathbb{R}^{C \times C}$ are the linear projection matrices. For simplicity, we ignore the output projection matrix $W^o$ and the normalization factor $d_k$, while also fixing the number of heads to 1.

The standard self-attention employs dense interaction for each query to gather crucial long-range dependencies. Hence, the computational complexity can be expressed as $O(2N^2C + 4NC^2)$, with $O(2N^2C)$ for dense interaction and $O(4NC^2)$ for linear projection.

*3.1.2 Local Self-Attention.* Since the computational complexity scales quadratically with the number of tokens, researchers are exploring the use of local self-attention, aiming to reduce computational complexity and introduce the local induction bias of convolution. Assuming the kernel size of the neighborhood sampling is $n$. For a flattened input feature map $x \in \mathbb{R}^{N \times C}$, the output of the $i$-th token $z_i$ after local self-attention can be expressed as:

$$(q, k, v) = (xW^q, xW^k, xW^v) \tag{3}$$

$$(\tilde{q}, \tilde{k}, \tilde{v}) = (\mathrm{Reshape}(q), \phi(k, n), \phi(v, n)) \tag{4}$$

$$z_i = \sum_{j=1}^{n^2} \frac{\exp(\tilde{q}_i \tilde{k}_j^{\mathrm{T}})}{\sum_{m=1}^{n^2} \exp(\tilde{q}_i \tilde{k}_m^{\mathrm{T}})} \tilde{v}_i \tag{5}$$

where $\tilde{q} \in \mathbb{R}^{N \times 1 \times C}$ denotes the queries after the *Reshape* operation. $\phi(\cdot, \cdot)$ is the sampling function, which can be easily accomplished by the built-in funcion *unfold* in PyTorch [28]. The $\tilde{k}, \tilde{v} \in \mathbb{R}^{N \times n^2 \times C}$ are the keys and values after neighborhood sampling, respectively.

The local self-attention restricts feature interaction to the local neighborhood of each query, thus the computational complexity

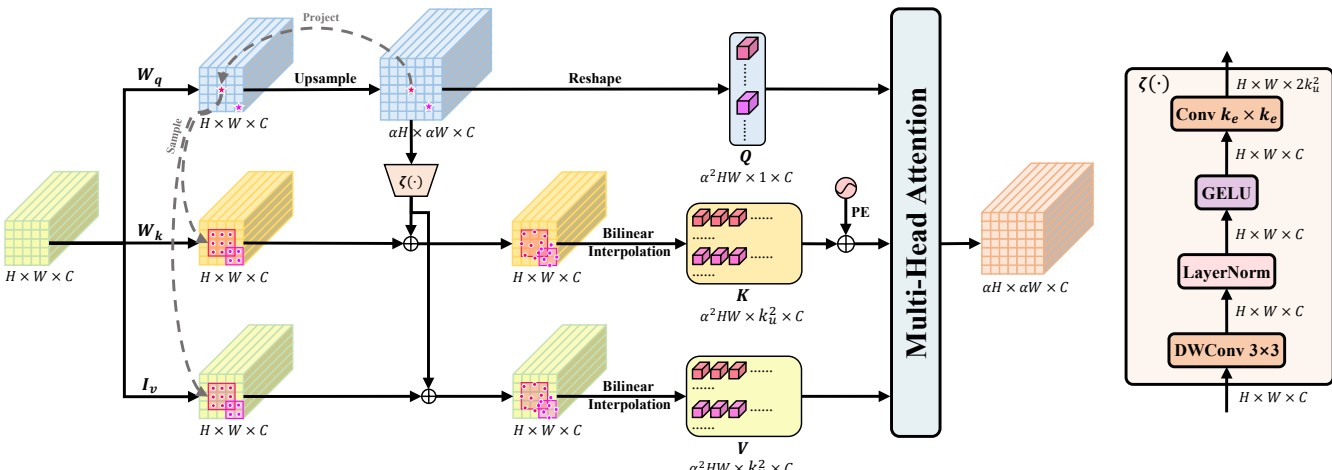

**Figure 4: The overall framework of LDA-AQU. Given an input feature map with size $H \times W \times C$ and an upsampling factor $\alpha$, LDA-AQU employs local deformable attention for feature upsampling, resulting in an output feature map with size $\alpha H \times \alpha W \times C$.**

can be expressed as $O(2n^2NC + 4NC^2)$, with $O(2n^2NC)$ for feature interaction and $O(4NC^2)$ for linear projection.

## 3.2 Feature Upsampling via Local Deformable Attention

*3.2.1 Extending Local Self-Attention for Upsampling.* Given an input feature map $X \in \mathbb{R}^{H \times W \times C}$ and upsampling factor $\alpha \in [1, +\infty]$, the output feature map $Y \in \mathbb{R}^{\alpha H \times \alpha W \times C}$ can be obtained through feature upsampling. Initially, the queries, keys, and values of the input feature map can be obtained through the linear mapping.

$$(Q, K, V) = (XW^Q, XW^K, XW^V) \tag{6}$$

Assuming the kernel size of neighborhood sampling is $k_u = 3$. As shown in Figure 3(b), let $p = (x, y)$ denote the coordinate of the point to be interpolated, where $x \in [0, W-1]$ and $y \in [0, H-1]$. Taking the grid arrangement format of aligned corner points as an example, we can obtain the corresponding coordinate $p' = (x', y')$ in the input feature map by

$$p' = \psi(p) = (x \frac{W}{\alpha W - 1}, y \frac{H}{\alpha H - 1}) \tag{7}$$

Let $r = \{(x_1, y_1), (x_2, y_1), ..., (x_3, y_3)\}$ denote the absolute coordinates of uniform neighboring points of $p'$. The upsampled result of point $p$ based on the local self-attention can be expressed as:

$$Y(p) = \sum_{s \in r} \frac{\exp(Q(p')K(s)^T)}{\sum_{t \in r} \exp(Q(p')K(t)^T)} V(s) \tag{8}$$

Let $F(p', s) = \frac{\exp(Q(p')K(s)^T)}{\sum_{t \in r} \exp(Q(p')K(t)^T)}$. When $W^V$ is an identity matrix, the above formula can be further simplified to

$$Y(p) = \sum_{s \in r} F(p', s) X(s) \tag{9}$$

Considering the properties of $Softmax$, we have $\sum_{s \in r} F(p', s) = 1$. Therefore, LA-AQU can be described as the adaptive acquisition of aggregation weights guided by the query features and the

reassembly of features from neighboring points. Indeed, many existing upsamplers can be formulated by upsampler based on local self-attention. We will delve into this concept in Section 3.3.

*3.2.2 Introducing Deformation Mechanism.* LA-AQU employs a fixed and uniform neighborhood sampling scheme for feature upsampling. Consequently, it becomes challenging to meet the upsampling needs across various complex scenarios simultaneously. As shown in Figure 2, solely aggregating the features of uniformly sampled neighboring points will result in the model overly emphasizing less significant background regions while neglecting the features of the object itself. Therefore, we further introduce the deformation mechanism and propose LDA-AQU.

The overall process of LDA-AQU is illustrated in Figure 4. Similarly, considering the input feature map $X \in \mathbb{R}^{H \times W \times C}$ and the output feature map $Y \in \mathbb{R}^{\alpha H \times \alpha W \times C}$, we initially obtain the $Q$, $K$, $V$ through Equation 6. To avoid extra dimensional transformations and reduce the number of parameters, we employ bilinear interpolation to upsample the matrix $Q \in \mathbb{R}^{H \times W \times C}$, resulting in $Q' \in \mathbb{R}^{\alpha H \times \alpha W \times C}$. Then, utilizing the built-in function *meshgrid* in PyTorch, we can generate the uniform coordinate matrix $P \in \mathbb{R}^{\alpha H \times \alpha W \times 1 \times 2}$ for the upsampled feature map. Through Equation 7, we can derive the reference point coordinate matrix $P'$ by projecting $P$ onto the input feature map.

Assuming the kernel size of neighborhood sampling is $k_u$, the initial offset $\Delta P$ of neighboring points can be obtained as

$$\Delta P = \{(-\lfloor k_u/2 \rfloor, -\lfloor k_u/2 \rfloor), ..., (\lfloor k_u/2 \rfloor, \lfloor k_u/2 \rfloor)\} \tag{10}$$

Subsequently, we can obtain the coordinate matrix of neighboring points $R \in \mathbb{R}^{\alpha H \times \alpha W \times k_u^2 \times 2}$ of the upsampled points through the broadcast mechanism by $R = P' + \Delta P$.

To enable the neighboring points to dynamically adjust their positions, we introduce a sub-network $\zeta(\cdot)$, utilizing query features to generate the query-guided sampling point offset matrix $\Delta R \in \mathbb{R}^{\alpha H \times \alpha W \times k_u^2 \times 2}$. Based on the uniform neighboring point coordinate matrix $R$ and the predicted offset matrix $\Delta R$, the final deformed

neighboring point coordinate matrix $R'$ can be obtained by

$$R' = R + \Delta R = R + \zeta(Q) \tag{11}$$

Finally, we utilize the point-wise features of matrix $Q'$ to interact with the corresponding local point features of $K, V$ to complete the upsampling task. The aforementioned process can be expressed as:

$$(\widetilde{Q}, \widetilde{K}, \widetilde{V}) = (\text{Reshape}(Q'), \Phi(K, R'), \Phi(V, R')) \tag{12}$$

$$Y = \text{Softmax}(\frac{\widetilde{Q}\widetilde{K}^{\text{T}}}{\sqrt{d_k}})\widetilde{V} \tag{13}$$

where $\widetilde{Q} \in \mathbb{R}^{\alpha H \times \alpha W \times 1 \times C}$ denotes the queries after *Reshape* operation. $\widetilde{K}, \widetilde{V} \in \mathbb{R}^{\alpha H \times \alpha W \times k_u^2 \times C}$ are the keys and values after neighboring sampling. To ensure differentiability, we use bilinear sampling function $\Phi(\cdot, \cdot)$ for sampling with non-integer offsets.

**Offset Predictor.** The detailed structure of the offset predictor $\zeta(\cdot)$, utilized to generate the sampling point offset matrix $\Delta R$, is depicted in Figure 4. We employ a $3 \times 3$ depthwise convolution to extend the perceptual range of the queries and utilize a convolutional layer with the kernel size of $k_e$ to predict the offset for $k_u^2$ points. The ablation studies of $k_e$ and $k_u$ can be found in Section 4.6.

**Offset Groups.** To enhance the model's capability in perceiving distinct channel features and adaptability across diverse scenarios, we partition the channels of the input feature maps within the offset predictor $\zeta(\cdot)$, employing varied offsets for different groups.

**Local Deformation Ranges.** To expedite convergence speed, we apply the *Tanh* function to the results of the Offset Predictor, limiting them to the range $[-1, 1]$. Additionally, we employ a factor $\theta$ to regulate the deformation range of the neighboring points. In our experiments, we observe that increasing the value of $\theta$ will bring some performance improvements. We believe that employing a larger local deformation range, combined with the grouping operation, will enable the model to capture a wider range of crucial features within the neighborhood for refining the upsampling results. More detailed analysis will be conducted in Section 4.6.

## 3.3 Relating to Other Upsampling Methods

In this section, we will explore the relationship between different upsamplers and LDA-AQU. Indeed, LA-AQU can already represent the majority of upsampling schemes.

**Bilinear Interpolation.** Similarly, let $X \in \mathbb{R}^{H \times W \times C}$ and $Y \in \mathbb{R}^{\alpha H \times \alpha W \times C}$ denote the input and output feature map, respectively. As shown in Figure 3(a), after projection via Equation 7, the feature vector of the upsampled point $p = (x, y)$ can be expressed as

$$Y(p) = \sum_{s \in r} F(p', s)X(s) \tag{14}$$

where $r = \{(x_1, y_1), (x_1, y_2), (x_2, y_1), (x_2, y_2)\}$ is the set of standard neighbor points of point $p'$. The bilinear interpolation kernel $F(p', s)$ in here can be represented as

$$F(p', s) = w(x', s_x)w(y', s_y) \tag{15}$$

where $w(a, b) = \max(0, 1 - |a - b|)$ represents the aggregation weight based on the distance between point pairs.

By comparing Equation 14 and Equation 9, we can conclude that upsampling through bilinear interpolation is indeed a special instance of LA-AQU. When the aggregate weight of points $\{(x_1, y_1), (x_2, y_1), (x_3, y_1), (x_1, y_2), (x_1, y_3)\}$ of $F(p', s)$ in Equation 9

equals zero, and the computation results are based on distance, LA-AQU will degrade into upsampling based on bilinear interpolation.

**CARAFE.** CARAFE and LA-AQU both compute dynamic content-aware aggregation weights for neighboring points of upsampled points. The distinction lies in LA-AQU using query-related dynamic features to interactively generate kernel weights, whereas CARAFE uses a convolutional layer. Moreover, the query-guided weight generation in LA-AQU enables the model to prioritize features that closely align with its own content.

**DySample.** DySample achieves feature upsampling through sampling. With the deformation mechanism, LDA-AQU achieves similar effects but employs the query-guided mechanism for neighboring deformation and feature aggregation. Compared to DySample, LDA-AQU leverages object-specific features effectively, and the kernel-based upsampling scheme aligns better with human intuition, *i.e.*, using neighboring points for feature inference.

It is noteworthy that LDA-AQU avoids using the PixelShuffle operator, in contrast to CARAFE and DySample, enabling it to achieve any desired multiple of feature upsampling.

## 3.4 Complexity Analysis

Given an input feature map with the shape of $H \times W \times C$, the overall computational complexity of LDA-AQU can be expressed as $O(2HWC^2 + 2\alpha^2 k_u^2 HWC + 2\alpha^2 k_u^2 k_e^2 HWC)$, with $O(2HWC^2)$ for linear projection, $O(2\alpha^2 k_u^2 HWC)$ for attention interaction, and $O(2\alpha^2 k_u^2 k_e^2 HWC)$ for deformed offsets prediction. Note that we ignore the computational complexity of depthwise convolution and positional encoding as they are considerably lower than the aforementioned blocks. To summarize, LDA-AQU exhibits linear computational complexity with the number of input tokens.

## 4 Experiments

## 4.1 Experimental Settings

We evaluate the effectiveness of proposed LDA-AQU on four challenging tasks, including object detection, instance segmentation, semantic segmentation, and panoptic segmentation.

**Datasets and Evaluation Metrics.** We utilize the challenging MS COCO dataset [19] to evaluate the effectiveness of LDA-AQU across object detection, instance segmentation, and panoptic segmentation tasks, respectively. For object detection and instance segmentaion tasks, we report the standard COCO metrics of Mean Average Precision (mAP). For the panoptic segmentation task, we report the PQ, SQ, and RQ metrics [15] as in Dysample [21]. For the semantic segmentation task, we conduct performance comparison using the ADE20K dataset [40] and report the Average Accuracy (aAcc), Mean IoU (mIoU), Mean Accuracy (mAcc) metrics.

**Implementation Details.** We evaluate the effectiveness of LDA-AQU using MMDetection [2] and MMSegmentation [5] toolboxes. Specifically, we adopt Faster R-CNN [31], Mask R-CNN [11], Panoptic FPN [14] and UperNet [36] as the baseline models. If unspecified, the offset groups, local deformation ranges, and channel size reduction factors are set to 2, 11, and 4, respectively. We use $1\times$ training schedule for object detection, instance segmentation and panoptic segmentation tasks. For semantic segmentation, the model is trained for 160K iterations. All other training strategies and hyperparameters remain the same as in [21] for fair comparison.

**Table 1: Performance comparison of Object Detection on MS COCO based on Faster R-CNN. * denotes that the channel reduction factor is set to 16 to balance FLOPs and performance. The best is highlighted in bold, and the second best is underlined.**

| Faster R-CNN | Backbone | $AP$ | $AP_{50}$ | $AP_{75}$ | $AP_S$ | $AP_M$ | $AP_L$ | Params | FLOPs | Reference |
|---|---|---|---|---|---|---|---|---|---|---|
| Nearest | ResNet-50 | 37.5 | 58.2 | 40.8 | 21.3 | 41.1 | 48.9 | 46.8M | 208.5G | - |
| Deconv | ResNet-50 | 37.3 | 57.8 | 40.3 | 21.3 | 41.1 | 48.0 | +2.4M | +12.6G | - |
| PS [32] | ResNet-50 | 37.5 | 58.5 | 40.4 | 21.5 | 41.5 | 48.3 | +9.4M | +50.2G | CVPR16 |
| CARAFE [35] | ResNet-50 | 38.6 | 59.9 | 42.2 | **23.3** | 42.2 | 49.7 | +0.3M | +1.6G | ICCV19 |
| IndexNet [24] | ResNet-50 | 37.6 | 58.4 | 40.9 | 21.5 | 41.3 | 49.2 | +8.4M | +46.4G | ICCV19 |
| A2U [6] | ResNet-50 | 37.3 | 58.7 | 40.0 | 21.7 | 41.1 | 48.5 | +38.9K | +0.3G | CVPR21 |
| FADE [25] | ResNet-50 | 38.5 | 59.6 | 41.8 | 23.1 | 42.2 | 49.3 | +0.2M | +3.4G | ECCV22 |
| SAPA-B [26] | ResNet-50 | 37.8 | 59.2 | 40.6 | 22.4 | 41.4 | 49.1 | +0.1M | +2.4G | NeurIPS22 |
| DySample [21] | ResNet-50 | 38.7 | 60.0 | 42.2 | 22.5 | 42.4 | **50.2** | +65.5K | +0.3G | ICCV23 |
| LDA-AQU* | ResNet-50 | 38.9 | 60.4 | 42.4 | **23.3** | 42.8 | 49.7 | +41.0K | +0.4G | - |
| LDA-AQU | ResNet-50 | **39.2** | **60.7** | **42.7** | 22.9 | **43.0** | 50.1 | +0.2M | +1.7G | - |

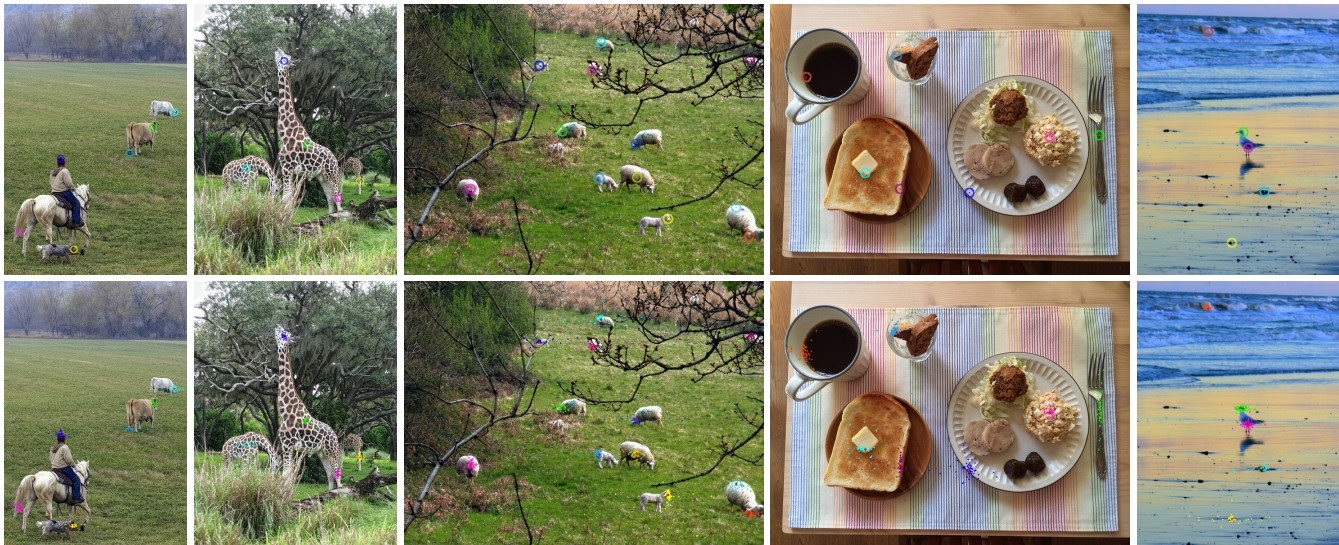

**Figure 5: Visualizations of some upsampled points (first row) and their deformed neighboring points (second row). Colored rings depict upsampled points, while scatter points of the same color are corresponding deformed neighboring points.**

## 4.2 Object Detection

As shown in Table 1, LDA-AQU outperforms the best model, DySample, by a large margin of 0.5 AP (39.2 AP v.s 38.7 AP). Moreover, LDA-AQU incurs only a minor increase in FLOPs and parameters, nearly matching CARAFE but surpassing it by 0.6 AP. For fair comparison, we adjust the channel size reduction factor to 16 to ensure a comparable FLOPs and parameters to DySample. Results in Table 1 indicate that the LDA-AQU maintains superior performance even with similar FLOPs and parameters (38.9 AP v.s 38.7 AP).

## 4.3 Instance Segmentation

As illustrated in Table 2, LDA-AQU improves the performance of the Mask RCNN by 1.5 bbox AP (39.8 AP v.s 38.3 AP) and 1.5 mask AP (36.2 AP v.s 34.7 AP), surpassing DySample by 0.2 bbox AP (39.8 AP v.s 39.6 AP) and 0.5 mask AP (36.2 AP v.s 35.7 AP). Considering the size of the input feature map in mask head is $14 \times 14$, we reduce

the local deformation range of LDA-AQU embedded into the mask head to 5 to prevent instability in the training process caused by excessive deformation range. The detailed ablation study of the deformation ranges on the mask head can be found in Section 4.6.

## 4.4 Panoptic Segmentation

As shown in Table 3, LDA-AQU surpasses all previous methods by a significant margin and maintains a similar number of parameters. For instance, LDA-AQU surpasses DySample by 0.7 PQ (42.2 PQ v.s 41.5 PQ). Even with a strong backbone like ResNet-101, LDA-AQU still achieves a PQ gain of 1.5 (43.7 PQ v.s 42.2 PQ), surpassing DySample by 0.7 PQ (43.7 PQ v.s 43.0 PQ).

## 4.5 Semantic Segmentation

As shown in Table 4, by replacing the upsamplers with LDA-AQU in the FPN and Multi-level Feature Fusion (FUSE) of UperNet, the

**Table 2: Performance comparison of Instance Segmentation on MS COCO based on Mask R-CNN.**

| Mask R-CNN | Task | Backbone | $AP$ | $AP_{50}$ | $AP_{75}$ | $AP_S$ | $AP_M$ | $AP_L$ |
|---|---|---|---|---|---|---|---|---|
| Nearest | Bbox | ResNet-50 | 38.3 | 58.7 | 42.0 | 21.9 | 41.8 | 50.2 |
| Deconv | | ResNet-50 | 37.9 | 58.5 | 41.0 | 22.0 | 41.6 | 49.0 |
| PS [32] | | ResNet-50 | 38.5 | 59.4 | 41.9 | 22.0 | 42.3 | 49.8 |
| CARAFE [35] | | ResNet-50 | 39.2 | 60.0 | 43.0 | 23.0 | 42.8 | 50.8 |
| IndexNet [24] | | ResNet-50 | 38.4 | 59.2 | 41.7 | 22.1 | 41.7 | 50.3 |
| A2U [6] | | ResNet-50 | 38.2 | 59.2 | 41.4 | 22.3 | 41.7 | 49.6 |
| FADE [25] | | ResNet-50 | 39.1 | 60.3 | 42.4 | 23.6 | 42.3 | 51.0 |
| SAPA-B [26] | | ResNet-50 | 38.7 | 59.7 | 42.2 | 23.1 | 41.8 | 49.9 |
| DySample [21] | | ResNet-50 | 39.6 | 60.4 | 43.5 | 23.4 | 42.9 | 51.7 |
| LDA-AQU | | ResNet-50 | 39.8 | 60.8 | 43.5 | 23.8 | 43.6 | 51.6 |
| Nearest | | ResNet-101 | 40.0 | 60.4 | 43.7 | 22.8 | 43.7 | 52.0 |
| DySample | | ResNet-101 | 41.0 | 61.9 | 44.9 | 24.3 | 45.0 | 53.5 |
| LDA-AQU | | ResNet-101 | 41.3 | 62.3 | 45.2 | 24.4 | 45.5 | 53.7 |
| Nearest | Segm | ResNet-50 | 34.7 | 55.8 | 37.2 | 16.1 | 37.3 | 50.8 |
| Deconv | | ResNet-50 | 34.5 | 55.5 | 36.8 | 16.4 | 37.0 | 49.5 |
| PS [32] | | ResNet-50 | 34.8 | 56.0 | 37.3 | 16.3 | 37.5 | 50.4 |
| CARAFE [35] | | ResNet-50 | 35.4 | 56.7 | 37.6 | 16.9 | 38.1 | 51.3 |
| IndexNet [24] | | ResNet-50 | 34.7 | 55.9 | 37.1 | 16.0 | 37.0 | 51.1 |
| A2U [6] | | ResNet-50 | 34.6 | 56.0 | 36.8 | 16.1 | 37.4 | 50.3 |
| FADE [25] | | ResNet-50 | 35.1 | 56.7 | 37.2 | 16.7 | 37.5 | 51.4 |
| SAPA-B [26] | | ResNet-50 | 35.1 | 56.5 | 37.4 | 16.7 | 37.6 | 50.6 |
| DySample [21] | | ResNet-50 | 35.7 | 57.3 | 38.2 | 17.3 | 38.2 | 51.8 |
| LDA-AQU | | ResNet-50 | 36.2 | 57.9 | 38.5 | 17.3 | 39.1 | 52.8 |
| Nearest | | ResNet-101 | 36.0 | 57.6 | 38.5 | 16.5 | 39.3 | 52.2 |
| DySample | | ResNet-101 | 36.8 | 58.7 | 39.5 | 17.5 | 40.0 | 53.8 |
| LDA-AQU | | ResNet-101 | 37.5 | 59.2 | 40.2 | 17.6 | 41.1 | 54.3 |

**Table 3: Performance comparison of Panoptic Segmentation on MS COCO based on Panoptic FPN.**

| Panoptic FPN | Backbone | $PQ$ | $PQ^{th}$ | $PQ^{st}$ | $SQ$ | $RQ$ | Params |
|---|---|---|---|---|---|---|---|
| Nearest | ResNet-50 | 40.2 | 47.8 | 28.9 | 77.8 | 49.3 | 46.0M |
| Deconv | ResNet-50 | 39.6 | 47.0 | 28.4 | 77.1 | 48.5 | +1.8M |
| PS [32] | ResNet-50 | 40.0 | 47.4 | 28.8 | 77.1 | 49.1 | +7.1M |
| CARAFE [35] | ResNet-50 | 40.8 | 47.7 | 30.4 | 78.2 | 50.0 | +0.2M |
| IndexNet [24] | ResNet-50 | 40.2 | 47.6 | 28.9 | 77.1 | 49.3 | +6.3M |
| A2U [6] | ResNet-50 | 40.1 | 47.6 | 28.7 | 77.3 | 48.0 | +29.2K |
| FADE [25] | ResNet-50 | 40.9 | 48.0 | 30.3 | 78.1 | 50.1 | +0.1M |
| SAPA-B [26] | ResNet-50 | 40.6 | 47.7 | 29.8 | 78.0 | 49.6 | +0.1M |
| DySample [21] | ResNet-50 | 41.5 | 48.5 | 30.8 | 78.3 | 50.7 | +49.2K |
| LDA-AQU | ResNet-50 | 42.2 | 48.7 | 32.4 | 78.6 | 51.5 | +0.1M |
| Nearest | ResNet-101 | 42.2 | 50.1 | 30.3 | 78.3 | 51.4 | 65.0M |
| CARAFE [35] | ResNet-101 | 42.8 | 49.7 | 32.5 | 79.1 | 52.1 | +0.2M |
| DySample [21] | ResNet-101 | 43.0 | 50.2 | 32.1 | 78.6 | 52.4 | +49.2K |
| LDA-AQU | ResNet-101 | 43.7 | 50.3 | 33.5 | 79.6 | 53.0 | +0.1M |

**Table 4: Performance comparison of Semantic Segmentation on ADE20K based on UperNet.**

| UperNet | Backbone | aAcc | mIoU | mAcc |
|---|---|---|---|---|
| Bilinear | ResNet-50 | 79.08 | 39.78 | 52.81 |
| PS [32] | ResNet-50 | 79.34 | 39.10 | 50.36 |
| CARAFE [35] | ResNet-50 | 79.45 | 41.0 | 52.59 |
| FADE [25] | ResNet-50 | 79.97 | 41.89 | 54.64 |
| SAPA-B [26] | ResNet-50 | 79.47 | 41.08 | 53.67 |
| DySample [21] | ResNet-50 | 79.82 | 41.08 | 53.00 |
| LDA-AQU | ResNet-50 | 80.11 | 42.31 | 55.37 |
| Bilinear | ResNet-101 | 80.27 | 42.52 | 54.91 |
| DySample [21] | ResNet-101 | 80.30 | 42.39 | 55.82 |
| LDA-AQU | ResNet-101 | 80.52 | 43.41 | 56.53 |

**Table 5: Performance comparison of various local deformation ranges in the FPN of Faster R-CNN.**

| $\theta$ | $AP$ | $AP_{50}$ | $AP_{75}$ | $AP_S$ | $AP_M$ | $AP_L$ |
|---|---|---|---|---|---|---|
| 5 | 38.7 | 60.3 | 42.0 | 23.0 | 42.3 | 49.8 |
| 7 | 39.0 | 60.5 | 42.5 | 22.9 | 42.6 | 50.1 |
| 9 | 39.1 | 60.6 | 42.3 | 23.0 | 43.0 | 49.9 |
| 11 | 39.2 | 60.7 | 42.7 | 22.9 | 43.0 | 50.1 |
| 13 | 39.0 | 60.4 | 42.4 | 22.8 | 43.0 | 50.3 |

**Table 6: Performance comparison of various local deformation ranges in the mask head of Mask R-CNN.**

| $\theta$ | Task | $AP$ | $AP_{50}$ | $AP_{75}$ | $AP_S$ | $AP_M$ | $AP_L$ |
|---|---|---|---|---|---|---|---|
| 3 | bbox | 39.6 | 60.6 | 43.2 | 23.7 | 42.9 | 51.1 |
| | segm | 36.0 | 57.6 | 38.4 | 17.3 | 38.4 | 52.7 |
| 5 | bbox | 39.7 | 60.8 | 43.5 | 23.8 | 43.6 | 51.2 |
| | segm | 36.2 | 57.9 | 38.5 | 17.3 | 39.1 | 52.8 |
| 7 | bbox | 39.4 | 60.5 | 43.0 | 23.1 | 43.2 | 50.8 |
| | segm | 36.0 | 57.4 | 38.5 | 17.1 | 38.8 | 52.4 |

mIoU of baseline model has been improved from 39.78 to 42.31, surpassing CARAFE by 1.31 mIoU and DySample by 1.23 mIoU. In addition, we also report the aAcc and mAcc metrics of the model. As depicted in Table 4, the performance of LDA-AQU remains superior on these two metrics compared to other upsamplers.

## 4.6 Ablation Study

We conduct ablation studies on MS COCO using Faster R-CNN and Mask R-CNN to verify the impact of hyperparameters in LDA-AQU.

**Local Deformation Ranges.** Initially, we assess the impact of $\theta$ in the FPN of Faster R-CNN. As shown in Table 5, setting $\theta$ of each LDA-AQU to 11 achieves optimal performance (39.2 AP). As $\theta$ decreases, the performance of the model gradually diminishes.

We believe the reason is that a few neighboring points are enough to cover sufficient information to ensure accurate interpolation. Therefore, the model prefers to find broader contextual information as auxiliary items to optimize the upsampling results.

Then, we verify the influence of $\theta$ in the mask head of Mask R-CNN. As shown in Table 6, when $\theta$ of mask head is set to 5, the model achieves the best performance. The reason is that the size of the input feature map of the mask head is $14 \times 14$, so using a larger $\theta$ will make it difficult for the model to focus on local details.

**Offset Groups.** We evaluate the impact of the offset groups on model performance. As shown in Table 7, the model achieves the optimal performance when the number of groups is set to 2. Excessive groupings will lead to a decreased size of features utilized for predicting offsets, thus impeding model learning.

**Channel Size Reduction Factors.** Finally, we evaluate the impact of channel reduction factors on model performance. As

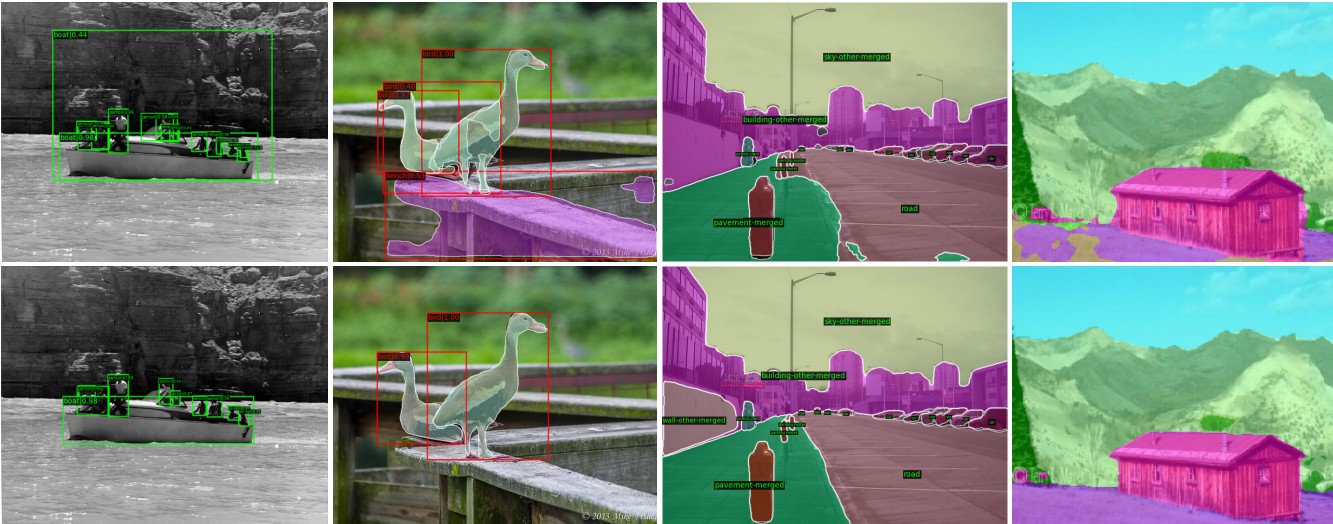

**Figure 6: Qualitative comparison between baseline models (first row) and LDA-AQU (second row) across various tasks (*i.e.,* object detection, instance segmentation, panoptic segmentation, and semantic segmentation, from left to right).**

**Table 7: Performance comparison of different offset groups.**

| Groups | AP | $AP_{50}$ | $AP_{75}$ | $AP_S$ | $AP_M$ | $AP_L$ |
|--------|------|------|------|------|------|------|
| 1 | 38.9 | 60.3 | 42.1 | 22.8 | 42.5 | **50.3** |
| 2 | **39.2** | **60.7** | **42.7** | 22.9 | **43.0** | 50.1 |
| 4 | 39.0 | 60.3 | 42.3 | 23.0 | 42.7 | **50.3** |
| 8 | 38.8 | 60.2 | 42.1 | **23.1** | 42.6 | 50.1 |

**Table 8: Performance comparison of various channel size reduction factors.**

| Factor | AP | $AP_{50}$ | $AP_{75}$ | $AP_S$ | $AP_M$ | $AP_L$ |
|--------|------|------|------|------|------|------|
| 16 | 38.9 | 60.4 | 42.4 | 23.3 | 42.8 | 49.7 |
| 8 | 38.9 | 60.3 | 42.2 | 23.3 | 42.8 | 49.7 |
| 4 | 39.2 | 60.7 | 42.7 | 22.9 | **43.0** | 50.1 |
| 2 | **39.4** | **60.8** | **42.8** | **23.7** | **43.0** | **50.4** |

illustrated in Table 8, when the reduction factor is set to 16, LDA-AQU yields an AP gain of 1.4 for Faster R-CNN (38.9 AP v.s 37.5 AP). By reducing the channel reduction factor, the performance of the model gradually improved, finally reaching 39.4 AP. In order to balance the performance and computational complexity of the model, we set the channel reduction factor of LDA-AQU to 4.

### 4.7 Visual Inspection and Analysis

In this section, we provide some visualizations and analysis of deformed neighbor points and results across different tasks. More visualizations can be found in the supplementary materials.

**Deformed Neighboring Points.** We visualize the locations of some upsampled points and their corresponding deformed neighboring points. As shown in Figure 5, LDA-AQU can adaptively adjust the position of neighbor points according to the query features

(*e.g.*, the fork in the fourth column). Even in occlusion scenes (*e.g.*, third column of Figure 5), LDA-AQU demonstrates a tendency to prioritize the featrues of object itself over the occluder (*i.e.*, branches).

**Qualitative Experiments.** We also conduct qualitative experiments to verify the effectiveness of LDA-AQU. Specifically, we visualize the the results of the baseline models (*e.g.*, Faster R-CNN, Mask R-CNN, *etc.* ) and LDA-AQU across various visual tasks. As shown in Figure 6, our model exhibits superior performance, thereby validating the effectiveness of LDA-AQU.

## 5 Conclusion

In this paper, we introduce local self-attention into the upsampling task. Compared with previous methods, the upsampling based on local self-attention (LA-AQU) naturally incorporates the feature guidance mechanism without necessitating high-resolution input. Additionally, to enhance the adaptability of LA-AQU to complex upsampling scenarios, we further introduce the query-guided deformation mechanism and propose LDA-AQU. Finally, LDA-AQU can dynamically adjust the location and aggregation weight of neighboring points based on the features of the upsampled point. Through extensive experiments on four dense prediction tasks, we evaluate the effectiveness of LDA-AQU. Specifically, LDA-AQU has consistently demonstrated leading performance across above tasks, while maintaining a comparable FLOPs and parameters. For future work, we intend to explore dynamic local deformation ranges and investigate additional application scenarios, including image restoration, image inpainting, downsampling, *etc.*

## Acknowledgments

This work was partially funded by the National Key Research and Development Plan of China (No. 2018AAA0101000) and the National Natural Science Foundation of China under grant 62076028.

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
