# OpenReview forum: "LDA-AQU: Adaptive Query-guided Upsampling via Local Deformable Attention"
_acmmm.org/ACMMM/2024/Conference — MM2024 Poster_

### Official Review · Reviewer_whWw · 2024-05-15

**Rating:** 4
**Confidence:** 3

**Summary:**

This paper introduces a novel dynamic kernel-based upsampler, LDA-AQU, which utilizes query features to guide the model in adaptively adjusting the positions and aggregation weights of neighboring points across various complex scenarios to meet the upsampling requirements. Experimental results demonstrate that LDA-AQU consistently outperforms previous state-of-the-art upsamplers in four dense prediction tasks and also achieves significant performance improvements.

**Strengths:**

1. The idea of incorating local self-attention into feature upsampling tasks is interesting and quite simple, and the subsequent proposed deformation mechanism bassed on queries and contexts can adjust the position to enhance the performance.
2. This paper has also shown the relationship between the proposed method and other well-known upsampling methods, which demonstrates the generality and dynamic scheme of LDA-AQU.
3. The experiments on different tasks consistently outperforms competing methods on all tested benchmarks. Several ablation studies are carried out to validate the functionality of hyperparameters.

**Limitations:**

1. Could the authors clarify the relationship between the factor $\theta$ and the offset range? Providing the formulation or equation would help illustrate this relationship.
2. Have the authors investigated the impact of the upsampling factor $\alpha$ on the performance of the proposed LDA-AQU? Conducting ablations on this factor would provide valuable insights.
3. It would be helpful to explain the rationale behind using $I_v$ instead of $W_v$ in the value branch depicted in Figure 4.
4. Which padding modes are employed when points are located on the image edge? Additionally, does this choice of padding modes affect the performance of LDA-AQU? Clarification on this aspect would be beneficial.

**Suitability:**

2

---

### Official Review · Reviewer_M46H · 2024-05-24

**Rating:** 5
**Confidence:** 3

**Summary:**

This paper finds existing upsamplers either lack specific feature guidance or necessitate the utilization of high-resolution feature maps. Hence, the authors propose a novel dynamic kernel-based upsampler, LDA-AQU, which introduces local self-attention and deformation mechanism into the upsampling task.  LDA-AQU utilizes the feature of queries to guide the model in adaptively adjusting the position and aggregation weight of neighboring points. LDA-AQU is also lightweight and can be easily integrated into various model architectures. LDA-AQU is evaluated across four dense prediction tasks: object detection, instance segmentation, panoptic segmentation, and semantic segmentation.

**Strengths:**

1. This paper proposes a novel dynamic upsampler which incorporates local self-attention into feature upsampling.
2. The proposed LDA-AQU has demonstrated effectiveness on various tasks and datasets.
3. The authors have provided comprehensive comparison between the proposed LDA-AQU and existing upsamplers.
4. The exposition of equations in the paper is remarkably clear, and the figures and visualizations are beautifully crafted.

**Limitations:**

1. LDA-AQU introduces self attention and deformable convolution into upsampling. Hence, LDA-AQU appears quite similar to existing deformable attention [1]. Please clarify the difference.
2. The authors adopt model parameters and FLOPS to evaluate the computational efficiency, which is misleading. For example, deformable convolution results in irregular neighbors, which makes the memory accessing pattern discontinuous and then increase the memory accessing time. However, FLOPS and parameters cannot adequately reflect such difference. Please provide execution latency and throughput for the evaluation in Figure 1 and Table 1.
3. The comparison in Figure 1 is confusing. Are the parameters and FLOPS shown in the figure for the entire network or only for the upsampler module? If it's the former, then is it ensured that the rest of the network remain consistent during the comparison?
4. The authors' experiments demonstrate that feature-guided upsampling achieves better performance across multiple tasks. However, I am curious about the underlying physical significance. Could an explanation be provided?
5. In line 532 "convolutional layer" rather than "convolutional alyer"

[1] Zhu X, Su W, Lu L, et al. Deformable DETR: Deformable Transformers for End-to-End Object Detection[C]//International Conference on Learning Representations. 2020.

Overall, I acknowledge the work presented in this paper, but further explanations and experiments are needed.

**Suitability:**

2

---

### Official Review · Reviewer_sXdr · 2024-05-25

**Rating:** 3
**Confidence:** 3

**Summary:**

This manuscript introduces a novel feature upsampling method called LDA-AQU, which leverages a local self-attention mechanism along with a query-guided deformation mechanism to adaptively adjust the positions and aggregation weights of neighboring points, thereby effectively enhancing the performance of feature upsampling in deep convolutional neural networks. In dense prediction tasks such as object detection, instance segmentation, panoptic segmentation, and semantic segmentation, LDA-AQU has demonstrated competitive performance while maintaining comparable computational load and parameter count.

**Strengths:**

The manuscript introduces an innovative upsampling method called LDA-AQU, which enhances the performance and flexibility of upsampling by integrating a local self-attention mechanism with a query-guided deformation mechanism. The authors have not only proposed a novel approach but also delved deeply into the principles behind it, including the incorporation of local self-attention and the design of the deformation mechanism. Additionally, LDA-AQU maintains a computational load and parameter count comparable to existing methods, suggesting that it offers high efficiency and scalability in practical applications.

**Limitations:**

1, In Figure 6, the authors demonstrate a comparison between LDA-AQU and a baseline approach, but I haven't come across a description of this baseline approach within the paper, which leaves me uncertain as to what it is being compared with.

2, In Section 3.2.2, the authors mention, "To avoid continuous projection and sampling, we employ bilinear interpolation to upsample the matrix...," but I don't understand the cause-and-effect relationship here; more explanation from the authors would aid the readers' comprehension.

3, In Tables 1, 2, 3, and 4, the authors present comparative results of LDA-AQU and other methods on various dense prediction tasks. I suggest that the authors could consider including more recent approaches in their comparison, such as "Learning To Zoom and Unzoom" from CVPR 2023.

4, Regarding the comparative methods, I've noticed that the experiments conducted by the authors were all based on convolutional networks. To substantiate the effectiveness of the method and further enhance the persuasiveness of the paper, I recommend that the authors also perform some comparative experiments on Transformer-type networks.

5, In the ablation study section, the authors have conducted detailed experiments on the values of various hyperparameters, but it seems that they have overlooked the ablation experiments on the module structure. Why is depthwise convolution (DWConv) used in some places while regular convolution is used in others? Why was bilinear interpolation chosen over other methods? How can it be determined that the current module design is optimal?

**Suitability:**

3

---

### Meta-Review · Area_Chair_iSHF · 2024-07-01

**Recommendation:** Accept (Poster)
**Confidence:** 5

**Metareview:**

The initial reviews were mixed and the reviewers raised questions about the implementation details. The authors' rebuttal addressed most of the reviewers' concerns. Now all reviewers lean towards the acceptance of the paper.